# Osteopontin, kidney injury molecule-1, and fetuin-A as prognostic markers of end-stage renal disease: A systematic review and meta-analysis

Yongki Welliam[1], Bendix Samarta Witarto[1], Visuddho Visuddho[1], Citrawati Dyah Kencono Wungu[2,3]*, Raden Mohammad Budiarto[4], Hendri Susilo[4,5], Chaq El Chaq Zamzam Multazam[6]

1 Faculty of Medicine, Medical Program, Universitas Airlangga, Surabaya, Indonesia, 2 Department of Physiology and Medical Biochemistry, Faculty of Medicine, Universitas Airlangga, Surabaya, Indonesia, 3 Institute of Tropical Disease, Universitas Airlangga, Surabaya, Indonesia, 4 Cardiology and Vascular Medicine Department, Dr. Soetomo General Hospital, Surabaya, Indonesia, 5 Department of Cardiology and Vascular Medicine, Universitas Airlangga Hospital, Surabaya, Indonesia, 6 National Heart and Lung Institute, Faculty of Medicine, Imperial College London, London, England

* citrawati.dyah@fk.unair.ac.id

## Abstract

### Background

The global prevalence of chronic kidney disease (CKD) as a major renal disease is increasing rapidly. The progression of CKD may lead to end-stage renal disease (ESRD). Current diagnostic and prognostic methods still have some limitations. This study aims to evaluate the potential and predictive ability of Osteopontin (OPN), Kidney injury molecule-1 (KIM-1), and Fetuin-A on the incidence of ESRD in CKD patients.

### Methods

A systematic review and meta-analysis were carried out based on the PRISMA guideline on registered databases for studies published up to December 21, 2023. The concentrations of each marker were then reported in pooled standardized mean difference (SMD) and hazard ratio (HR). Subgroup analysis was performed based on age, location, and KIM-1 specimen.

### Results

We included 21 studies involving 15,983 patients. Meta-analysis revealed that increasing OPN (SMD = 5.52, 95% CI = 1.59–9.44, p = 0.01) and KIM-1 (SMD = 1.45, 95% CI = 0.50–2.39, p = 0.0027), as well as decreasing Fetuin-A level (SMD = -1.31, 95% CI = -2.37 − -0.26, p = 0.01) were significant in CKD patients with ESRD. Chronic kidney disease patients with increased KIM-1 levels showed 1.13 times increased risk of ESRD (HR = 1.13, 95% CI = 1.10–1.17, p <0.0001). Subgroup analysis showed that increased KIM-1 in urine or blood was strongly associated with ESRD, and decreased Fetuin-A levels in Asians had a significant association with the incidence of ESRD.

**Data availability statement:** All relevant data are within the manuscript and its Supporting Information files.

**Funding:** The author(s) received no specific funding for this work.

**Competing interests:** The authors have declared that no competing interests exist.

## Conclusion

Osteopontin, KIM-1, and Fetuin-A significantly reflect ESRD in CKD patients, making them potential prognostic indicators.

## Introduction

Chronic kidney disease (CKD) is a structural and functional abnormality of the kidneys lasting three months or more which is characterized by a decrease in the glomerular filtration rate of less than 60 ml/minute/1.73 m2 of body surface area [1]. In 2017, the global prevalence of CKD reached 843.6 million individuals (about 11.1% of the world population), with a mortality rate raised to 1.2 million [2,3]. Individual comorbidities, such as diabetes mellitus and hypertension, are the primary cause of the increased prevalence and mortality rate of CKD [4]. When kidney function has declined to the point where the kidneys can no longer maintain adequate filtration and excretion of waste products, CKD can progress to End-stage Renal Disease (ESRD). According to KDIGO, ESRD is defined as kidney failure characterized by a GFR < 15 ml/min/1.73 m2 or requiring dialysis therapy or other kidney replacement therapy [5]. Although the incidence of end-stage renal disease (ESRD) is currently relatively low (1–2%), the progression of CKD might lead to it [6]. However, it still requires more attention because the prevalence of ESRD in the United States has been estimated to be more than 500,000 populations [7]. The ESRD mortality rate in Taiwan is also quite high, reaching 11.8% [8]. Even though dialysis therapy has now been widely applied as a treatment for CKD, data shows that the mortality of ESRD patients still reaches 20–50% [7].

Nowadays, many diagnostic methods for CKD have been developed, such as the estimation of glomerular filtration rate (GFR), proteinuria, radiological and histopathological examinations, and other renal markers. Current diagnostic techniques, such as GFR, serum creatinine, and albuminuria, have limited ability to measure the progression and worsening of clinical outcomes among patients with CKD. Moreover, the use of these conventional markers appears to be insufficiently sensitive and specific in describing the progression of early-stage disease. Furthermore, the use of eGFR or serum creatinine is also inadequate in reflecting the presence of intrarenal injury or signaling pathway disorders [9]. By being able to detect kidney function decline earlier, we can improve therapy outcomes and patient prognosis. Therefore, physicians need to be able to develop a predictive method since the risk of systemic morbidity in CKD patients has been observed to be higher than in the general population [10]. Research on biomarkers which play a role in kidney pathological processes is now starting to be widely explored. Some of these markers can include renal function markers and vascular calcification markers such as OPN, KIM-1, and fetuin-A.

The hyperfiltration occurring in CKD patients leads to the activation of the RAAS system, including angiotensin II, which stimulates OPN production in the glomerulus. Elevated levels of OPN result in biological changes and contribute to renal fibrosis [11]. The presence of injury and inflammation in the renal system can also be represented by an increase in KIM-1 levels due to the lysis of renal tubular cells [12]. Fetuin-A is a protein that serves various functions in inhibiting pathological processes such as vascular calcification and kidney stones. A cohort study by Caglar *et al*. [13] revealed that a decrease in Fetuin-A levels correlates with an increase in the severity of chronic kidney failure as it inhibits vascular calcium-phosphate deposition by forming fetuin-mineral complexes, which in turn diminish the progression of end stage CKD [14]. The diversity of results obtained from various studies prompted this systematic review and meta-analysis to provide more concrete and accurate results regarding the ability of the three markers to estimate the risk of ESRD in CKD patients. As an implication,

findings from this study could be provided as a basis for daily clinical practice utilization in the future.

## Methods

This systematic review and meta-analysis were prepared by adopting the flow set out in Preferred Reporting Items for Systematic Reviews and Meta-Analysis (PRISMA) 2020 [15]. To avoid the risk of duplication of studies, the study protocol has been registered on the International Prospective Register of Systematic Reviews (PROSPERO) (CRD42022357894). The research was carried out involving three reviewers (YOW, BSW, and VIV) in the process from literature search to data extraction. If there are discrepancies in the selection of included studies, discussions involving a third author will be held to reach a consensus.

### Search strategy

The search was conducted on the Pubmed, Scopus, Web of Science, Sciencedirect, EBSCO/ CINAHL, ProQuest, and preprints (Bioxriv and Medrxiv) which contain the required data using Medical Subject Heading Terms (MeSH) in the keywords of "Osteopontin" or "OPN" or "Sialoprotein 1" or "Secreted Phosphoprotein 1" or "Uropontin", "KIM-1" or "Kidney injury molecule-1" or "TIMD1" or "hepatitis A virus receptor" or "HAV cellular receptor" or "TIM -1", "Fetuin-A" or "AHSG" or "α2-Heremans Schmid glycoprotein", "end stage renal disease" or "ESRD" or "stage V" or "stage 5" or "end stage kidney disease" or "end stage kidney failure" or "end stage renal failure and "chronic kidney diseases" or "chronic renal insufficiency" or "chronic renal diseases" or "chronic renal failure" or "chronic kidney disease" published up to January 5, 2023. The search for studies was conducted using Boolean operators, specifically "AND" and "OR". Further research strategy is summarized in S1 Table **in** S1 File.

### Study selection

The initial screening and selection were based on the titles and abstracts of articles. Screening was then continued by reviewing the entire content of the articles if the full-text articles were obtained. Inclusion criteria in this study were: 1) adult patients (≥ 18 years); 2) studies comparing the osteopontin, KIM-1, and/or fetuin-A levels between non-ESRD and ESRD in CKD patients; 3) cohort, case-control, or cross-sectional study design; 4) studies with Mean and SD data or data that supports the calculation of the Mean and SD, or studies containing Hazard Ratio (HR) values and its 95% Confidence Intervals; and 5) available in full-text. Articles will be excluded if they meet the following criteria: 1) case series study design, case report, review, or clinical trial; 2) studies that did not provide complete data; 3) duplicated studies.

### Quality assessment

Quality assessment of the studies was carried out using the Newcastle Ottawa Scale (NOS), which is a method used in analyzing the quality of observational studies (cohort, case-control, or cross-sectional) included in systematic reviews and/or meta-analysis [16,17].

### Data extraction

Data from each study were extracted and then arranged in a data extraction table. The collected data included the first author's name, publication year, sample size, age of the study population, study location, concentrations of OPN, KIM-1, Fetuin-A markers, examination sample, the number of CKD patients without ESRD and with ESRD, study design, mean and standard deviation, or hazard ratio.

## Data analysis

Meta-analysis was performed by collecting pooled standardized mean difference (SMD) and pooled HR data from each study. Standardized mean difference is a statistical conclusion in meta-analysis that is used to compare the means of two groups that have been measured on different scales or manner [18]. The effect of the marker level changes toward ESRD occurrence is reflected as a standard mean difference. Standard mean difference values between 0.2–0.5 are considered to have a small effect, 0.5–0.8 have a moderate effect, and more than 0.8 have a strong/large effect [19].

Meanwhile, the hazard ratio describes how many times an event occurs compared to the comparison group within a certain period [20]. In this study, analysis using HR were performed to estimate the risk of change in the biomarker value towards the incidence of ESRD in CKD patients. In each included study, the mean and standard deviation values of each marker were extracted to calculate the SMD. In studies that did not include both values, alternative data were used, which includes median (m), upper limit (a), and lower limit of marker concentration (b) to determine the mean and standard deviation values by Hozo et al. formula [21]. The extracted hazard ratio value was the adjusted Hazard Ratio (aHR), meaning the HR that has been adjusted by removing various covariates and analyzed based on a significance value of $p < 0.05$ and CI 95%.

Heterogeneity between studies was evaluated using the Q-test and I2 test. The random effect model was used if the I2 test value was > 50% and the Q-test was significant ($p < 0.1$) [22]. Outside this range, the fixed effect model was used. Statistical analysis was carried out using Review Manager 5.4 (The Cochrane Collaboration, Oxford, UK) and STATA version 16.0 software (Stata Corporation, College Station, TX, USA).

Subgroup analysis was carried out according to age and study location for all biomarkers, and sample specimen for the KIM-1 particularly. Assessment of publications was carried out quantitatively using Egger's test if it involves ≥ 10 studies. Publication bias was indicated if the Egger's test results showed $p < 0.05$. Sensitivity analysis was carried out by excluding studies with a risk of bias or moderate to low quality (NOS QS score < 7). The SMD and HR values were then reevaluated to determine the consistency of the results after studies with a risk of bias were excluded.

## Results

### Search and selection results

Study searches were conducted on five large databases, including PubMed, Scopus, Web of Science, CINAH, and Proquest, with a total of 653 studies. After checking for duplicates and screening for title and abstract, several studies were excluded due to inappropriate population, irrelevant methods, different markers and outcomes, no outcome values, and non-English article, leaving 21 studies included. Study searches were also conducted on other sources as gray literature, including Medrxiv (9 studies) and Biorxiv (24 studies). However, no studies were included in this additional search (Fig 1).

### Study characteristics

The characteristics of the studies included in this meta-analysis are summarized in Table 1. A total of 21 studies involving three markers (OPN, KIM-1, and Fetuin-A) with two types of measurements of end-stage renal disease incidence are presented. In total, there were 15,983 patients included in this clinical outcome study. Based on its marker, there were three studies using the OPN marker, nine studies with KIM-1, and 11 studies investigating fetuin-A. Two studies (Schmidt et al. and

**PRISMA 2020 flow diagram for new systematic reviews which included searches of databases, registers and other sources**

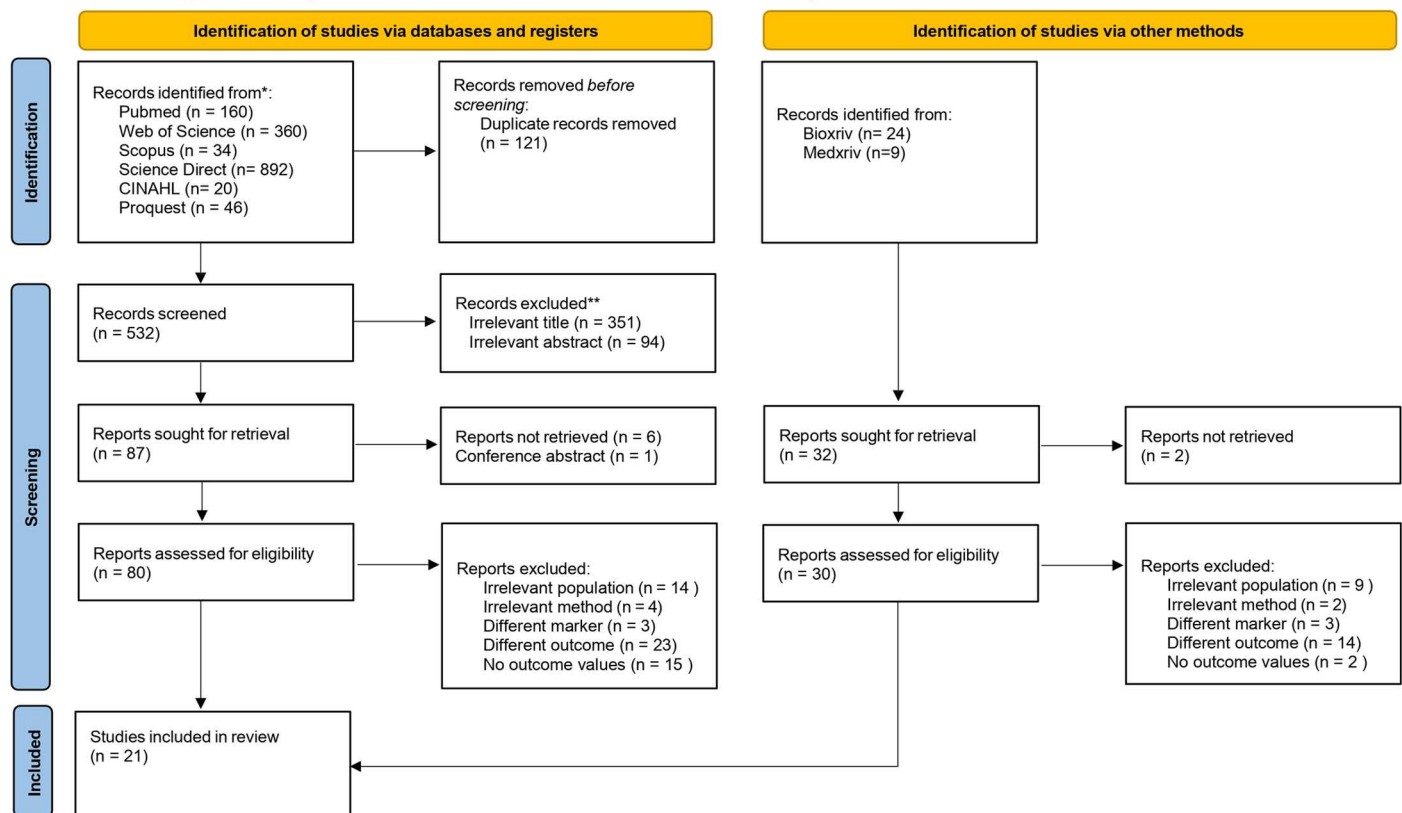

*Consider, if feasible to do so, reporting the number of records identified from each database or register searched (rather than the total number across all databases/registers).
**If automation tools were used, indicate how many records were excluded by a human and how many were excluded by automation tools.

*From:* Page MJ, McKenzie JE, Bossuyt PM, Boutron I, Hoffmann TC, Mulrow CD, et al. The PRISMA 2020 statement: an updated guideline for reporting systematic reviews. BMJ 2021;372:n71. doi: 10.1136/bmj.n71. For more information, visit: http://www.prisma-statement.org/

**Fig 1. PRISMA Flow Chart 2020.**

Gluba-Brzózka et al.) [23,24] involved more than one marker. Based on the type of specimen (either blood or urine), analysis of KIM-1 is then carried out with various measurement units. Based on the type of measurement, there are 16 studies where the standard mean difference value can be calculated and nine studies where the hazard ratio (HR) is measured. Of these 21 studies, 12 of them were studies that had a mean patient age ≥ 60 years. Groups of CKD patients with specific comorbidity were only reported in two studies, which involved diabetes mellitus patients [25,26]. The population in this study is also quite diverse with population distribution, Asia (5 studies), Europe (6 studies), America (6 studies), Australia (1 study), and Africa (3 studies).

## Quality assessment

The results of the study quality assessment of the various included studies are summarized in S2-**4 Table in** S1 File. Study quality assessment is differentiated according to the design of each included study (cohort, case-control, or cross-sectional) using the Newcastle Ottawa Scale Quality Assessment [16,17]. The results show that 9 studies had good quality (score 7–9). In case-control studies, only 1 out of 4 studies had a high risk of bias. In the cross-sectional study, 6 studies were categorized as " good studies", while 2 studies were "satisfactory studies". The quality assessment of this study is based on 3 assessment domains according to the study design.

**Table 1. Characteristics of included studies.**

| Author (Year) | Study Design | Study Location | Population Ages* | Total Sample (female) | Sample Size | | Comorbid | Marker | Specimen Type | Outcome Value ESRD | Outcome Value Non – ESRD | Measurement Unit | *Hazard Ratio (HR) (95% CI)* |
|---|---|---|---|---|---|---|---|---|---|---|---|---|---|
| | | | | | ESRD | Non-ESRD | | | Mean ± SD* | Mean ± SD* | | | |
| Schrauben *et al.* (2021) [26] | Prospective Cohort | United States | 61.0 ± 9.3 | 1315 (562) | 538 | 777 | Diabetes | KIM-1 | Plasma | – | – | – | 1.17 (1.05-1.3) |
| Vassallo *et al.* (2019)[27] | Prospective Cohort | United States | 71.45 ± 2.95 | 112 [38] | 13 | 99 | N/A | KIM-1 | Plasma | 339.23 ± 81.88 | 284.95 ± 55.14 | ng/L | 1.12 (0.9-1.39) |
| Maharem *et al.* (2013) [28] | Case-control | Egypt | 47.67 ± 11.65 | 57 [34] | 40 | 17 | N/A | Fetuin-A | Serum | 0.26 ± 0.14 | 0.17 ± 0.04 | g/L | – |
| Kamińska *et al.* (2021) [29] | Case-control | Poland | 61.67 ± 15.72 | 57 [27] | 38 | 19 | N/A | OPN | Serum | 38 ± 5.14 | 27.76 ± 4.06 | ng/mL | 1.42 (0.74-2.7) |
| Sigrist *et al.* (2009) [30] | Prospective Cohort | United Kingdom | 60 ± 14 | 134 (70) | 88 | 46 | N/A | Fetuin-A | Serum | 0.27 ± 0.21 | 0.25 ± 0.13 | ng/mL | |
| Alderson, 2017 [31] | Prospective Cohort | Great Britain | 64 ± 14 | 458 (175) | 135 | 323 | N/A | KIM-1 | Blood | – | – | – | 1.23 (0.95-1.6) |
| Azoz *et al.* (2022) [25] | Case-control | Egypt | 57.54 ± 7.24 | 70 [20] | 35 | 35 | Diabetes Mellitus | OPN | Serum | 168.42 ± 7.56 | 71.91 ± 4.28 | ng/mL | – |
| Schmidt *et al.* (2022) [23] | Prospective Cohort | United States | 52.8 ± 16.6 | 524 (267) | 124 | 400 | N/A | KIM-1 | Plasma | – | – | – | 1.19 (1.03-1.38) |
| | | United States | 57.7 ± 11.0 | 3800 (1694) | 1153 | 2647 | N/A | KIM-1 | Plasma | – | – | – | 1.1 (1.06-1.15) |
| Caglar *et al.* (2008) [13] | Cross sectional | Turkey | 44.0 ± 12.4 | 198 (96) | 46 | 152 | N/A | Fetuin-A | Serum | 0.25 ± 0.02 | 0.32 ± 0.04 | g/L | – |
| Smith *et al.* (2013) [32] | Cross sectional | Australia | 66.55 ± 15.34 | 65 | 54 | 11 | N/A | Fetuin-A | Serum | 0.17 ± 0.05 | 0.22 ± 0.03 | g/L | – |
| Mutluay *et al.* (2019) [33] | Cross sectional | Turkey | 53 ± 11.43 | 238 (98) | 131 | 107 | N/A | Fetuin-A | Serum | 10.31 ± 1.09 | 66.39 ± 11.22 | ng/mL | – |
| Mikami *et al.* (2008) [34] | Cross sectional | Japan | 63.63 ± 11.65 | 85[24] | 16 | 69 | N/A | Fetuin-A | Serum | 0.94 ± 0.57 | 1.05 ± 0.7 | g/L | – |
| Alderson *et al.* (2016) [35] | Prospective Cohort | Great Britain | 64.5 ± 14.7 | 1982 (747) | 266 | 1716 | N/A | KIM-1 | Plasma | 482.66 ± 67.16 | 333.87 ± 80.47 | pg/mL | – |
| Kim *et al.* (2013) [36] | Case-control | Korea | – | 151 | 81 | 70 | N/A | Fetuin-A | Serum | 117.2 ± 2.69 | 141.43 ± 10.21 | µg/mL | – |
| Can *et al.* (2021) [37] | Cross sectional | Turkey | 46.8 ± 10.3 | 92 [49] | 8 | 84 | N/A | Fetuin-A | Serum | 77.7 ± 20.2 | 88.27 ± 21.19 | ng/mL | – |
| Dubin *et al.* (2018) [38] | Prospective Cohort | United States | 71 ± 9.3 | 1472 (677) | – | – | N/A | KIM-1 | Urine | – | – | – | 1.24 (1.08-1.42) |
| Hsu *et al.* (2016) [39] | Prospective Cohort | United States | 59.5 ± 10.8 | 2466 (1131) | – | – | N/A | KIM-1 | Urine | – | – | – | 1.16 (1.05-1.28) |
| Malhotra *et al.* (2020) [40] | Prospective Cohort | United States | 73 ± 9 | 2428 (979) | – | – | N/A | KIM-1 | Urine | – | – | – | 2.34 (1.31-6.17) |
| Zeidan *et al.* (2012) [41] | Cross sectional | Egypt | 40.25 ± 10.8 | 60 [29] | 12 | 48 | N/A | Fetuin-A | Serum | 0.19 ± 0.11 | 0.24 ± 0.08 | g/L | – |
| Gluba-Brzózka *et al.* (2016) [24] | Cross sectional | Poland | 67.2 ± 11.7 | 80 [35] | 20 | 60 | N/A | Fetuin-A | Blood | 125.2 ± 63.3 | 110.93 ± 77.18 | ng/ mL | – |
| | Cross sectional | Poland | 67.2 ± 11.7 | 80 [35] | 20 | 60 | N/A | OPN | Blood | 28.7 ± 19.7 | 23.6 ± 27.14 | ng/ml | – |
| Gluba-Brzózka *et al.* (2014) [42] | Cross sectional | Poland | 67.1 ± 12.9 | 139 (77) | 49 | 90 | N/A | Fetuin-A | Blood | 30.1 ± 2.68 | 50.9 ± 24.58 | ng/mL | – |

*Presentation of continuous data uses mean ± standard deviation

**CI:** *Confidence Interval*; **ESRD:** *End-stage Renal Disease*; **KIM-1:** *Kidney Injury Molecule-1*; **OPN:** *Osteopontin*; **SD:** *Standard Deviation*

Regarding the OPN marker, only three studies with a total of 207 samples were included, and the results showed that increasing OPN levels had a significant effect on the incidence of ESRD in CKD patients (SMD = 5.52, 95% CI = 1.59–9.44, p = 0.01). Heterogeneity in this analysis was categorized as high, therefore, a random effect model was used ($I^2$ = 98.55%) (Fig 2A). For the KIM-1 marker, only two studies were included with a total of 2,094 samples. The results showed that increasing KIM-1 levels had a large and significant effect on the incidence of ESRD in CKD patients (SMD = 1.45, 95% CI = 0.50–2.39, p = 0.0027) (Fig 2B). Measurement of the effect of Fetuin-A levels on ESRD was assessed in 11 studies and showed that decreasing Fetuin-A levels had a significantly large effect on the incidence of ESRD in patients with chronic renal failure (SMD = -1.31, 95% CI = -2.37 – (-0.26), p = 0.01) (Fig 2C). In contrast to the two previous outcomes, Fetuin-A has sufficient studies to carry out subgroup analysis based on age and geographic location. The results show that in the groups aged ≥ 60 years and <60 years, neither of them had a significant effect on the incidence of ESRD in CKD patients respectively (≥ 60 years: SMD = -0.38, 95% CI = -0.93–0.18, p = 0.18; <60 years: SMD = -1.94, 95% CI = -4.32–0.43, p = 0.11) **(S1 and 2 Fig in** S1 File). Subgroup analysis based on geographic location showed that only studies in Asia had significance on the incidence of ESRD (SMD = -2.56, 95% CI = -4.57 – (-0.55), p = 0.01), while studies located on the continents of Europe, Africa, and Australia were not **(S3-6 Fig in** S1 File). Overall analysis showed that geographic location did not have a significant effect on decreasing Fetuin-A levels towards the incidence of ESRD in CKD patients (SMD = -1.31, 95% CI = -2.37 – (0.26), p = 0.09) (Table 2).

Meta-analysis of the OPN marker (Fig 3A) on ESRD outcomes in CKD patients only presented one study [29]. The analysis could not be carried out because the minimum number of studies in a meta-analysis is two studies [43]. For the KIM-1 marker, there were nine studies included with the results of the analysis showing that the incidence of ESRD was found to increase significantly 1.13 times in individuals who had increased KIM-1 levels (HR = 1.13, 95% CI = 1.10–1.17, p = <0.0001) (Fig 3B). The fixed-effect model was used in this analysis because of its low level of heterogeneity ($I^2$ = 8.24%). Subgroup analysis was performed on this group based on the type of specimen used. It turned out that both increased urine and blood KIM-1 concentrations significantly increased the risk of cardiovascular disease in chronic renal failure patients (urine KIM-1 HR = 1.19, 95% CI = 1.09–1.28, p = < 0.0001; blood HR KIM-1 = 1.12, 95% CI = 1.08–1.16, p = <0.00001) **(S7 and 8 Fig in** S1 File). Unfortunately, the risk of ESRD in CKD patients could not be measured using the Fetuin-A marker in this study due to study number limitations.

## Publication bias assessment and sensitivity analysis

Assessment of publication bias was performed quantitatively (Egger's test), especially in the study groups which included more than 10 studies. In this study, the authors intend to analyze the publication bias in two outcome groups with an adequate number of studies: (1) the group that analyzed changes in Fetuin-A concentrations on the incidence of ESRD with the effect of SMD measurements and (2) a group that analyzed changes in KIM-1 concentration on the risk of ESRD with the effect of measuring the hazard ratio. In the first group, the Egger's test showed an insignificant value, which means there is no potential bias in the included studies (Z = -1.05; p = 0.2920). However, in the second group, the risk of publication bias was found to be significant obtained from the results of the Egger's test (p = 0.01).

Sensitivity analysis was done and following result was then obtained in Tables 3 and 4. Of the 12 clinical outcomes assessed, 5 were reanalyzed because they included these studies. Sensitivity analysis showed that there were no significant changes in the study group before and after excluding studies with risk of bias, with the notable exception of OPN which the SMD

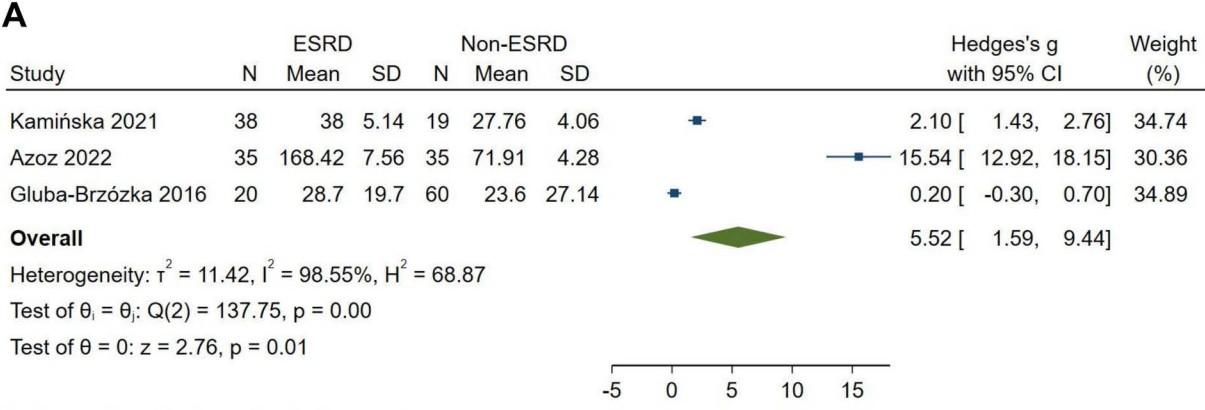

**A**

| Study | N | ESRD Mean | SD | N | Non-ESRD Mean | SD | | Hedges's g with 95% CI | Weight (%) |
|---|---|---|---|---|---|---|---|---|---|
| Kamińska 2021 | 38 | 38 | 5.14 | 19 | 27.76 | 4.06 | | 2.10 [ 1.43, 2.76] | 34.74 |
| Azoz 2022 | 35 | 168.42 | 7.56 | 35 | 71.91 | 4.28 | | 15.54 [ 12.92, 18.15] | 30.36 |
| Gluba-Brzózka 2016 | 20 | 28.7 | 19.7 | 60 | 23.6 | 27.14 | | 0.20 [ -0.30, 0.70] | 34.89 |
| **Overall** | | | | | | | | 5.52 [ 1.59, 9.44] | |

Heterogeneity: $\tau^2 = 11.42$, $I^2 = 98.55\%$, $H^2 = 68.87$

Test of $\theta_i = \theta_j$: Q(2) = 137.75, p = 0.00

Test of $\theta = 0$: z = 2.76, p = 0.01

Random-effects DerSimonian-Laird model

**B**

| Study | N | Treatment Mean | SD | N | Control Mean | SD | | Hedges's g with 95% CI | Weight (%) |
|---|---|---|---|---|---|---|---|---|---|
| Vassalo 2019 | 13 | 339.23 | 81.88 | 99 | 284.96 | 55.14 | | 0.92 [ 0.33, 1.51] | 45.50 |
| Alderson 2016 | 266 | 482.66 | 67.17 | 1,716 | 333.88 | 80.48 | | 1.89 [ 1.74, 2.03] | 54.50 |
| **Overall** | | | | | | | | 1.45 [ 0.50, 2.39] | |

Heterogeneity: $\tau^2 = 0.42$, $I^2 = 89.87\%$, $H^2 = 9.87$

Test of $\theta_i = \theta_j$: Q(1) = 9.87, p = 0.00

Test of $\theta = 0$: z = 3.00, p = 0.00

Random-effects DerSimonian-Laird model

**C**

| Study | N | ESRD Mean | SD | N | Non-ESRD Mean | SD | | Hedges's g with 95% CI | Weight (%) |
|---|---|---|---|---|---|---|---|---|---|
| Maharem 2013 | 40 | 0.26 | 0.14 | 17 | 0.17 | 0.04 | | 0.74 [ 0.16, 1.32] | 9.07 |
| Sigrist 2009 | 88 | 0.27 | 0.21 | 46 | 0.25 | 0.13 | | 0.11 [ -0.25, 0.46] | 9.22 |
| Caglar 2008 | 46 | 0.25 | 0.02 | 152 | 0.32 | 0.04 | | -1.92 [ -2.29, -1.54] | 9.21 |
| Smith 2013 | 54 | 0.17 | 0.05 | 11 | 0.22 | 0.04 | | -1.02 [ -1.68, -0.35] | 8.99 |
| Mutluay 2019 | 131 | 10.31 | 1.09 | 107 | 66.39 | 11.23 | | -7.39 [ -8.10, -6.67] | 8.94 |
| Mikami 2008 | 16 | 0.94 | 0.57 | 69 | 1.06 | 0.7 | | -0.18 [ -0.71, 0.36] | 9.10 |
| Kim 2013 | 81 | 0.12 | 0.002 | 70 | 0.14 | 0.01 | | -2.86 [ -3.31, -2.41] | 9.16 |
| Can 2021 | 8 | 77.7 | 20.2 | 84 | 88.27 | 21.19 | | -0.50 [ -1.22, 0.23] | 8.93 |
| Zeidan 2012 | 12 | 0.19 | 0.11 | 48 | 0.25 | 0.08 | | -0.68 [ -1.32, -0.05] | 9.02 |
| Gluba-Brzózka 2016 | 20 | 125.2 | 63.3 | 60 | 110.93 | 77.18 | | 0.19 [ -0.31, 0.69] | 9.13 |
| Gluba-Brzózka 2014 | 49 | 30.1 | 2.68 | 90 | 50.9 | 24.58 | | -1.04 [ -1.41, -0.67] | 9.22 |
| **Overall** | | | | | | | | -1.31 [ -2.37, -0.26] | |

Heterogeneity: $\tau^2 = 3.12$, $I^2 = 98.00\%$, $H^2 = 50.10$

Test of $\theta_i = \theta_j$: Q(10) = 501.02, p = 0.00

Test of $\theta = 0$: z = -2.44, p = 0.01

Random-effects DerSimonian-Laird model

**Fig 2. Forest plot for the pooled SMD in ESRD and non-ESRD among CKD patients.** (A) OPN; (B) KIM-1; and (C) Fetuin-A.

**Table 2. Summary of subgroup analysis results.**

| Marker | Outcome | Subgroup Analysis Variable | Measurement Effect | 95% CI | p-value |
|---|---|---|---|---|---|
| KIM-1 | ESRD | KIM-1 (Urine) | HR = 1.19 | 1.19–1.28 | <0.0001* |
| | | KIM-1 (Blood) | HR =1.12 | 1.08–1.16 | <0.0001* |
| Fetuin-A | ESRD | Age < 60 years old | SMD = -1.94 | (-4.32) – 0.43 | 0.11 |
| | | Age ≥ 60 years old | SMD = -0.38 | (-0.93) – 0.18 | 0.18 |
| | | Continent (Asian) | SMD = -2.56 | (-4.57) – (-0.55) | 0.01* |
| | | Continent (European) | SMD = -0.26 | (-1.07) – 0.55 | 0.54 |
| | | Continent (African) | SMD = 0.03 | (-1.36) – 1.43 | 0.96 |
| | | Continent (Australian) | SMD = -1.02 | (-1.68) – (-0.35) | N/A |
| | | Continent (American) | N/A | N/A | N/A |

* $p < 0.05$ means significant results.

increased remarkably from 5.52 to 15.54. This shows that the analysis results obtained have quite strong results.

## Discussion

Meta-analysis using the standard mean difference measurement effect was used to describe differences in changes in marker levels on the incidence of ESRD in CKD. Standard mean difference values between 0.2–0.5 are considered to have slightly effect, 0.5–0.8 have a moderate effect, and more than 0.8 have a strong/large effect [19]. This meta-analysis showed that increased levels of OPN and KIM-1 as well as decreased level of Fetuin-A have significant and large effects on the incidence of ESRD in CKD patients based on SMD calculations. In this study, analysis using HR were also performed to estimate the incidence of ESRD in CKD patients. The results showed that kidney injury molecule-1 is capable of predicting ESRD at 1.13 times higher in CKD patients.

The findings of the three markers in terms of end-stage renal disease incidence show the potential of each substance as a prognostic marker in patients with chronic renal failure. The results demonstrate that the marker OPN has a large and significant effect on the incidence of ESRD in patients with chronic renal failure. However, the prognostic power of OPN in describing the likelihood of ESRD has not been able to be evaluated statistically due to study limitations. The only existing study by Kamińska et al., [29] states that increased OPN concentrations have sufficient prognostic power with an increased risk of ESRD by 42% (HR=1.42) compared to normal. Unfortunately, subgroup analysis cannot be performed on this outcome due to the lack of included studies.

The potential of OPN as a prognostic marker of ESRD is explained by the following mechanisms. The fluid and salt imbalance that occurs in CKD patients due to decreased kidney function, causes increased activity of the RAAS system in the kidneys. Increasing this system will stimulate OPN production in the glomerulus. Elevated OPN levels will cause the accumulation of fibrotic tissue and mesangial matrix in the renal parenchyma, which subsequently leads to ESRD. This is due to the OPN alteration on the biological function of mesangial cells and podocytes through the secretion of TGF-β and MCP-1 [11]. In addition, increased OPN levels also correlate with increased chronic inflammation in patients with CKD [44]. This is in line with previous findings, which show that OPN levels are directly proportional to inflammatory markers such as CRP and IL-6 [45].

The role of KIM-1 as a prognostic marker for ESRD shows a large and significant effect in this meta-analysis. Increasing KIM-1 levels showed a significant increase in the incidence of

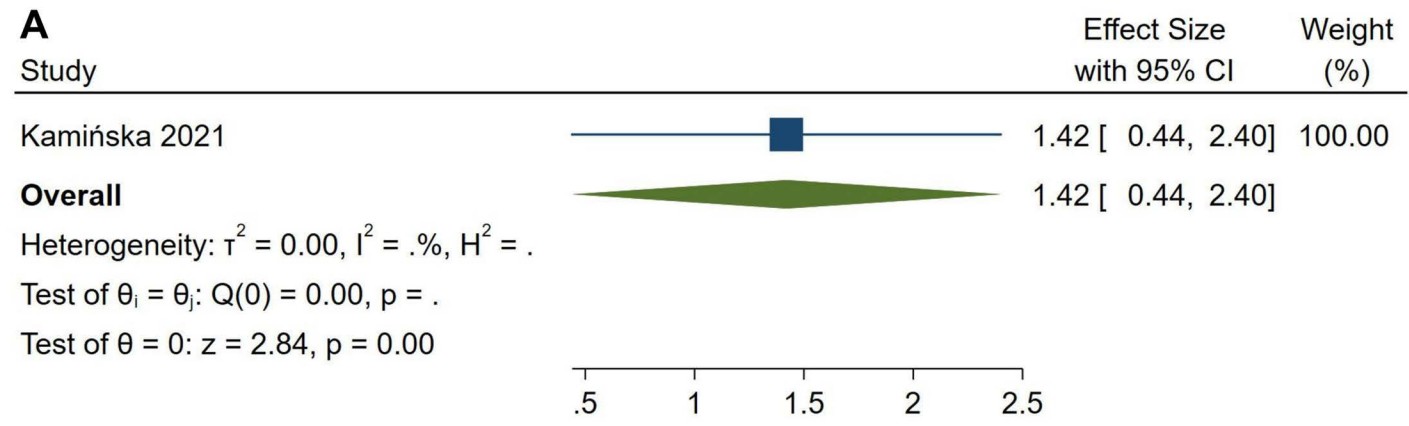

**Fig 3. Forest plot for the pooled HR in ESRD and non-ESRD among CKD patients.** (A) OPN and (B) KIM-1.

ESRD by 1.13%. A previous study by Alderson et al., [46] also showed similar finding. KIM-1 is a protein located in tubular epithelial cells, which plays a role in mediating phagocytosis of cells undergoing apoptosis or necrosis by binding to phosphatidylserine and oxidized lipid epitopes on the surface of cells undergoing apoptosis [47]. Through this mechanism, KIM-1

**Table 3. Summary of the findings results for ESRD in CKD patients before sensitivity analysis, along with the results of Egger's test calculation.**

| Measurement Effect | Marker | Total Study | SMD | 95% CI | I² | P value | Egger's test | Z score |
|---|---|---|---|---|---|---|---|---|
| SMD | OPN | 3 | 5.52 | 1.59–9.44 | 98.55% | 0.01* | <0.001** | 10.31 |
| | KIM-1 | 2 | 1.45 | 0.50–2.39 | 89.87% | 0.0027* | 0.0017** | -3.14 |
| | Fetuin-A | 11 | -1.31 | -2.37 – (-0.26) | 98.00% | 0.01* | 0.2920 | -1.05 |
| HR | KIM-1 | 9 | 1.13 | 1.10–1.17 | 8.24% | <0.0001* | 0.0100** | 2.57 |

*p < 0.05 means significant results.

**p < 0.05 means significantly bias results.

**Table 4. Sensitivity analysis results on the three prognostic markers.**

| Outcome | Group | Total Study | SMD | 95% CI | I² | p-value |
|---|---|---|---|---|---|---|
| SMD | OPN | 1 | 15.54 | 12.92–18.15 | – | <0.0001* |
| | KIM-1 | 2 | 1.45 | 0.50–2.39 | 89.87% | 0.0027* |
| | Fetuin-A | 9 | -1.61 | (-2.85) – (-0.37) | 98.25% | 0.01* |
| HR | KIM-1 | 9 | 1.13 | 1.10–1.17 | 8.24% | <0.0001* |

*p < 0.05 means significant results

can reflect worsening kidney function in patients with CKD. In the various extracted studies, the KIM-1 marker was obtained from 2 specimens (urine and blood). The prognostic power of each specimen was then reevaluated and compared. The results showed that both urine and blood KIM-1 were able to describe the incidence of ESRD significantly (p<0.05). Previous meta-analysis by Zhou et al., [48] also showed that urinary KIM-1 could significantly predict the incidence of ESRD. Its concentration in urine reflects direct release from damaged tubular cells, making it a specific biomarker for localized kidney injury. Existing studies even show more prognostic performance of urine KIM-1 than blood KIM-1. However, blood KIM-1 is also known to have the ability to reflect the progressive decline in eGFR and the risk of ESRD. In blood, elevated KIM-1 levels reflect more extensive kidney damage and potential involvement of systemic inflammation or the release of biomarkers from damaged kidneys into circulation [49]. Additionally, increased KIM-1 was also associated with increased mortality rates in CKD patients [38]. In conditions of renal tubular injury such as ESRD, tubular cell polarity is lost so that KIM-1 in the tubular cells can be released both into the tubular lumen and the interstitial fluid. The increase in transepithelial permeability after injury encourages the return of tubular contents into circulation [50]. Although the results of this meta-analysis are supported by existing studies, significant bias was detected in the included studies; thus, additional research with a larger number of studies may be required.

The ability of Fetuin-A to predict the possibility of ESRD is the most widely evaluated compared to OPN and KIM-1. The results showed that decreasing Fetuin-A had a large and significant effect on ESRD. However, the prognostic ability of Fetuin-A against ESRD could not be tested due to limited studies reporting HR values. Fetuin-A generally plays a role in inhibiting the process of vascular calcification, while its mechanism for worsening kidney disease is not fully understood. Toprak et al. [51] revealed that Fetuin-A acts as an acute phase reactant and its expression is inversely proportional to the degree of inflammation. This is also supported by the finding that Fetuin-A levels are inversely related to several pro-inflammatory cytokines such as IL-1b and CRP [52]. A cohort study by Caglar et al. [13] revealed that a decrease in Fetuin-A levels correlates with an increase in the severity of chronic kidney failure. Uremic toxicity, as one of the causes of worsening chronic kidney failure, is influenced by conditions such

as hypoalbuminemia, inflammation, and vascular calcification (VC), which are associated with a decrease in Fetuin-A levels [28]. Fetuin-A has both anti-inflammatory and pro-inflammatory actions. In conditions such as sepsis, injury, endotoxemia, and autoimmune diseases, the anti-inflammatory properties of Fetuin-A are activated [53]. Fetuin-A reduces inflammation by acting as an opsonin for cationic spermine, which regulates the innate immune response within immune cells [54]. However, during persistent inflammation, such as in chronic kidney disease, Fetuin-A levels tend to decrease [55]. Therefore, decreased Fetuin-A actually describes the occurrence of ESRD through the underlying inflammatory process.

Further analysis was carried out using subgroup analysis on the variables of age and geographic location. The results indicate that in both populations aged ≥ 60 years and < 60 years, neither showed significant changes between the decrease in Fetuin-A marker levels and the increase in ESRD incidence. This could imply two things: (1) the age groups used may not be suitable as subgroups in this analysis, resulting in non-significant results; (2) significant results may be obtained in other age groups. Consideration of subgroup analysis of the age variable was carried out, considering that the expression of this protein decreases with age [56]. Further research using lower age limits or diverse age groups may be needed to obtain more valid results. Subgroup analysis of geographic location variables showed insignificant results for the European, African, and Australian continents. The significant results in the Asian continent can be related to the high rate of CKD in Asia, reaching 434.3 million population with 65.6 million severe cases [57]. The etiology of CKD can vary depending on the location of various Asian countries. Some countries show different risk factors for CKD, such as infection incidence, socioeconomic life, lifestyle, pollution, and so on [58,59]. Analysis by region is important to determine priority areas for therapy and disease control through promotive and preventive efforts.

Based on our literature search, there has not been a systematic review and meta-analysis comprehensively addressing the prognostic value of OPN, KIM-1, and Fetuin-A concerning ESRD occurrence in CKD patients within a single study. Previous studies have also not thoroughly explored the potential use of these markers in relation to clinical outcomes in ESRD. This finding supports the potential use of all three markers as prognostic markers for ESRD in CKD patients through time series examination. Furthermore, this examination also aids in detecting renal injury and other intrarenal pathway abnormalities compared to using conventional examination methods.

Although this systematic review and meta-analysis have been conducted as meticulously as possible, the study still has several limitations. First, the three markers used have varying numbers of studies in this meta-analysis. Two study groups were also found to lack a sufficient number of studies for meta-analysis, specifically in assessing the prognostic strength of (1) OPN and (2) Fetuin-A on ESRD outcomes through the calculation of pooled Hazard Ratio (HR). Second, the study has not been able to identify threshold values/cut-offs for marker levels that may indicate potential risks for ESRD in CKD. Third, the high heterogeneity in this study may be caused by several factors. Variability in the characteristics of the included population such as age, gender, and geographical conditions impacts the distribution of marker concentration data unevenly. Additionally, most of the population actually has their own respective comorbidities that are difficult to control, which may influence the obtained results. The three study designs used in this research also allow for the emergence of heterogeneity due to various different analyses performed. Furthermore, the concentration of each marker may be measured by different instruments and units. Fourth, although most studies have good quality (NOS ≥7), some studies are still reported to have potential biases. Therefore, we attempted to reevaluate the study results by conducting sensitivity analyses, excluding studies with potential biases (NOS <7).

## Conclusion

Increased levels of OPN and KIM-1 have shown their significant and substantial effects on the occurrence of ESRD in CKD patients. Furthermore, increased KIM-1 level is also capable of predicting ESRD at 1.13 times higher in CKD patients based on the calculation of pooled Hazard Ratio. Whereas a decrease in Fetuin-A levels is also significantly associated with the occurrence of ESRD. This effect was larger particularly in the Asian population. As a conclusion, these findings establish OPN, KIM-1, and Fetuin-A as potential prognostic biomarkers of ESRD in patient with CKD. Further research is highly suggested to support the findings obtained from this study.

## Supporting information

**S1 File. Supplementary materials.**
(ZIP)

**S1 Checklist. PRISMA 2020 checklist.**
(DOCX)

**S1 Data. Extraction data table.**
(XLSX)

**S2 Data. Included and excluded studies.**
(XLSX)

## Author contributions

**Conceptualization:** Yongki Welliam, Bendix Samarta Witarto, Visuddho Visuddho, Citrawati Dyah Kencono Wungu.

**Data curation:** Yongki Welliam.

**Formal analysis:** Yongki Welliam.

**Investigation:** Yongki Welliam, Bendix Samarta Witarto, Visuddho Visuddho.

**Methodology:** Yongki Welliam, Bendix Samarta Witarto, Visuddho Visuddho, Citrawati Dyah Kencono Wungu.

**Project administration:** Yongki Welliam, Citrawati Dyah Kencono Wungu.

**Resources:** Yongki Welliam, Bendix Samarta Witarto, Visuddho Visuddho.

**Software:** Yongki Welliam, Bendix Samarta Witarto.

**Supervision:** Citrawati Dyah Kencono Wungu, Hendri Susilo, Raden Mohammad Budiarto.

**Validation:** Citrawati Dyah Kencono Wungu, Hendri Susilo, Raden Mohammad Budiarto, Chaq El Chaq Zamzam Multazam.

**Visualization:** Yongki Welliam.

**Writing – original draft:** Yongki Welliam, Bendix Samarta Witarto, Visuddho Visuddho.

**Writing – review & editing:** Citrawati Dyah Kencono Wungu, Hendri Susilo, Raden Mohammad Budiarto, Chaq El Chaq Zamzam Multazam.

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
