## [Decision Letter · Decision Letter 0]

16 Oct 2024

PONE-D-24-34763OSTEOPONTIN, KIDNEY INJURY MOLECULE-1, AND FETUIN-A AS PROGNOSTIC MARKERS OF END-STAGE RENAL DISEASE: A SYSTEMATIC REVIEW AND META-ANALYSISPLOS ONE

Dear Dr. Wungu,

Thank you for submitting your manuscript to PLOS ONE. After careful consideration, we feel that it has merit but does not fully meet PLOS ONE’s publication criteria as it currently stands. Therefore, we invite you to submit a revised version of the manuscript that addresses the points raised during the review process.

We look forward to receiving your revised manuscript.

Kind regards,

Wisit Kaewput, MD

Academic Editor

PLOS ONE

Journal Requirements: When submitting your revision, we need you to address these additional requirements. 1. Please ensure that your manuscript meets PLOS ONE's style requirements, including those for file naming. The PLOS ONE style templates can be found at https://journals.plos.org/plosone/s/file?id=wjVg/PLOSOne_formatting_sample_main_body.pdf and https://journals.plos.org/plosone/s/file?id=ba62/PLOSOne_formatting_sample_title_authors_affiliations.pdf 2. As required by our policy on Data Availability, please ensure your manuscript or supplementary information includes the following:  A numbered table of all studies identified in the literature search, including those that were excluded from the analyses.   For every excluded study, the table should list the reason(s) for exclusion.   If any of the included studies are unpublished, include a link (URL) to the primary source or detailed information about how the content can be accessed.  A table of all data extracted from the primary research sources for the systematic review and/or meta-analysis. The table must include the following information for each study:  Name of data extractors and date of data extraction  Confirmation that the study was eligible to be included in the review.   All data extracted from each study for the reported systematic review and/or meta-analysis that would be needed to replicate your analyses.  If data or supporting information were obtained from another source (e.g. correspondence with the author of the original research article), please provide the source of data and dates on which the data/information were obtained by your research group.  If applicable for your analysis, a table showing the completed risk of bias and quality/certainty assessments for each study or outcome.  Please ensure this is provided for each domain or parameter assessed. For example, if you used the Cochrane risk-of-bias tool for randomized trials, provide answers to each of the signalling questions for each study. If you used GRADE to assess certainty of evidence, provide judgements about each of the quality of evidence factor. This should be provided for each outcome.   An explanation of how missing data were handled.  This information can be included in the main text, supplementary information, or relevant data repository. Please note that providing these underlying data is a requirement for publication in this journal, and if these data are not provided your manuscript might be rejected. 3. Please review your reference list to ensure that it is complete and correct. If you have cited papers that have been retracted, please include the rationale for doing so in the manuscript text, or remove these references and replace them with relevant current references. Any changes to the reference list should be mentioned in the rebuttal letter that accompanies your revised manuscript. If you need to cite a retracted article, indicate the article’s retracted status in the References list and also include a citation and full reference for the retraction notice.

Reviewers' comments:

Reviewer's Responses to Questions

**Comments to the Author**

1. Is the manuscript technically sound, and do the data support the conclusions?

Reviewer #1: Yes

Reviewer #2: Yes

Reviewer #3: Partly

Reviewer #4: Yes

2. Has the statistical analysis been performed appropriately and rigorously? 

Reviewer #1: Yes

Reviewer #2: Yes

Reviewer #3: Yes

Reviewer #4: Yes

3. Have the authors made all data underlying the findings in their manuscript fully available?

Reviewer #1: Yes

Reviewer #2: Yes

Reviewer #3: Yes

Reviewer #4: Yes

4. Is the manuscript presented in an intelligible fashion and written in standard English?

Reviewer #1: Yes

Reviewer #2: Yes

Reviewer #3: Yes

Reviewer #4: Yes

5. Review Comments to the Author

Reviewer #1: The manuscript by Welliam et al. “Osteopontin, kidney injury molecule-1, and Fetuin-A as prognostic markers of end-stage renal disease: a systematic review and meta-analysis” looks interesting. This article aims to evaluate the potential and predictive ability of Osteopontin (OPN), Kidney injury molecule-1 (KIM-1), and Fetuin-A on the incidence of ESRD in CKD patients. This review and Meta-analysis showed that the increased KIM-1 in urine or blood was strongly associated with ESRD, and decreased Fetuin-A levels in Asians had a significant association with the incidence of ESRD. The authors concluded that Osteopontin, KIM-1, and Fetuin-A significantly reflect ESRD in CKD patients, making them potential prognostic indicators. The manuscript is well written and provides enough strategies to fulfill its aims, though language may need some polishing for better flow. In conclusion, I support publication of this work and believe that it would be interesting for researchers working in the field.

Reviewer #2: This manuscript written thoroughly and vigorously strengthens the review's question. Reviewer assist your manuscript based on PRISMA 2020 Expanded checklist, and most of the points are all elaborated in your manuscript. However, some minor revisions are needed, please access the pdf file attached.

Reviewer #3: This is a remarkable paper. There are a few minor points that need to be checked.

1.The effects of diabetes and hypertension were not taken into account: This study did not take into account the effects of comorbidities such as diabetes and hypertension on the progression of CKD and the onset of ESRD. Patients with comorbidities may have different conditions, so the fact that this was not adjusted for could be a source of bias.

2.The effectiveness of subgroup analysis by age and geographical location: The subgroup analysis examines the distribution of results by age and geographical location, but the age groups may not be appropriate (e.g., broad categories such as “under 60” and “over 60”). In addition, the geographical location analysis does not consider differences between countries or cultural and social backgrounds, and does not adequately explain why significant results were obtained only in the Asian region.

3.The effect of using different types of samples: Some of the included studies measured the concentrations of OPN, KIM-1 and fetuin-A using different samples, such as urine or blood, which may have caused the variation in results. The concentrations of markers in urine and blood may reflect different biological processes, and this must be taken into account in the analysis.

4.Uncertainty over cut-off values for clinical application: There is a problem with clinical application because there is no clear cut-off value for predicting the risk of developing ESRD based on the level of each biomarker. The cut-off value for biomarkers is an important indicator for predicting prognosis, so this point needs to be further investigated.

Reviewer #4: The manuscript titled "OSTEOPONTIN, KIDNEY INJURY MOLECULE-1, AND FETUIN-A AS PROGNOSTIC

MARKERS OF END-STAGE RENAL DISEASE: A SYSTEMATIC REVIEW AND META-ANALYSIS" by Welliam et al., is clearly written and their claims are fully supported by well-designed study and proper statistical analysis.

So, I recommend the publication of this manuscript in the esteemed journal PLOS ONE.

6. PLOS authors have the option to publish the peer review history of their article (what does this mean? ). If published, this will include your full peer review and any attached files.

**Do you want your identity to be public for this peer review?** For information about this choice, including consent withdrawal, please see our Privacy Policy .

Reviewer #1: No

Reviewer #2: No

Reviewer #3: **Yes: ** Taiki Ogasa

Reviewer #4: No

---

## [Author Response · Author response to Decision Letter 1]

16 Jan 2025

Point-by-point responses to Reviewers’ comments

We sincerely thank the reviewers for their valuable feedback and the Editorial Team of PLOS ONE for the opportunity to revise our manuscript. This response letter outlines the corrections made based on the reviewers’ comments to enhance clarity and improve the presentation of our findings. We hope our revised manuscript provides valuable insights into the OPN, KIM-1, and Fetuin- as prognostic markers of end-stage renal disease. Thank you for considering our work.

Reviewer 1

1. The manuscript by Welliam et al. “Osteopontin, kidney injury molecule-1, and Fetuin-A as prognostic markers of end-stage renal disease: a systematic review and meta-analysis” looks interesting. This article aims to evaluate the potential and predictive ability of Osteopontin (OPN), Kidney injury molecule-1 (KIM-1), and Fetuin-A on the incidence of ESRD in CKD patients. This review and Meta-analysis showed that the increased KIM-1 in urine or blood was strongly associated with ESRD, and decreased Fetuin-A levels in Asians had a significant association with the incidence of ESRD. The authors concluded that Osteopontin, KIM-1, and Fetuin-A significantly reflect ESRD in CKD patients, making them potential prognostic indicators. The manuscript is well written and provides enough strategies to fulfill its aims, though language may need some polishing for better flow. In conclusion, I support publication of this work and believe that it would be interesting for researchers working in the field.

R : We thank the reviewer for the positive feedbacks.

Reviewer 2

1. This manuscript written thoroughly and vigorously strengthens the review's question. Reviewer assist your manuscript based on PRISMA 2020 Expanded checklist, and most of the points are all elaborated in your manuscript. However, some minor revisions are needed, please access the pdf file attached.

R : We thank the reviewer for the suggestion. We have stated the date on when the data search was conducted as well as our financial support for this manuscript in Revised Manuscript file.

Reviewer 3

1. The effects of diabetes and hypertension were not taken into account: This study did not take into account the effects of comorbidities such as diabetes and hypertension on the progression of CKD and the onset of ESRD. Patients with comorbidities may have different conditions, so the fact that this was not adjusted for could be a source of bias.

R : We thank the reviewer for the suggestion. We agree that these comorbidities can affect study outcomes. However, our study primarily aimed to evaluate the potential and predictive ability of Osteopontin (OPN), Kidney injury molecule-1 (KIM-1), and Fetuin-A on the incidence of ESRD in CKD patients without stratifying for comorbid conditions to maintain the broader applicability of the findings. Moreover, while diabetes and hypertension are significant contributors to CKD progression, there are additional underlying factors such as genetic predisposition, dietary habits, medication adherence, or socioeconomic status that also play critical roles and were not adjusted for in this study. These factors can introduce variability in CKD progression independent of the presence of diabetes or hypertension.

To mitigate these potential biases, our methodology involved some adjustment or statistical methods including the use of random-effects models and sensitivity analyses. These approaches account for heterogeneity across studies, including variations in the prevalence of comorbidities like diabetes. We acknowledge this as a limitation and recommend further studies incorporating detailed stratification or multivariate adjustment for comorbid conditions are warranted to further elucidate their specific contributions.

2. The effectiveness of subgroup analysis by age and geographical location: The subgroup analysis examines the distribution of results by age and geographical location, but the age groups may not be appropriate (e.g., broad categories such as “under 60” and “over 60”). In addition, the geographical location analysis does not consider differences between countries or cultural and social backgrounds, and does not adequately explain why significant results were obtained only in the Asian region.

R : We thank the reviewer for the suggestion. The age categories ("under 60" and "over 60") were chosen to reflect a general distinction between younger and older populations, which is often used in studies related to chronic diseases, including CKD. This cutoff aligns with established clinical observations where individuals over 60 years old generally have a higher prevalence of CKD due to age-related decline in kidney function and increased comorbidities, as supported by studies like Coresh et al., 2007. We acknowledge that narrower age groups could provide more granular insights, but such stratification was limited by the data reported in the included studies. The significant results observed in the Asian region may be attributable to higher prevalence rates of CKD risk factors such as diabetes and hypertension in certain Asian populations and it has been already stated in our discussion (Hill et al., 2016). Additionally, genetic predispositions, dietary patterns, and healthcare access likely contribute to these findings. While we recognize that cultural and social differences between countries can influence outcomes, the meta-analysis aggregated data at the regional level due to inconsistent reporting of country-specific details in the included studies.

Reference :

• Hill, N.R., Fatoba, S.T., Oke, J.L., Hirst, J.A., O’Callaghan, C.A., Lasserson, D.S. and Hobbs, F.R., 2016. Global prevalence of chronic kidney disease–a systematic review and meta-analysis. PloS one, 11(7), p.e0158765.

• Coresh, J., Selvin, E., Stevens, L.A., Manzi, J., Kusek, J.W., Eggers, P., Van Lente, F. and Levey, A.S., 2007. Prevalence of chronic kidney disease in the United States. Jama, 298(17), pp.2038-2047.

3. The effect of using different types of samples: Some of the included studies measured the concentrations of OPN, KIM-1 and fetuin-A using different samples, such as urine or blood, which may have caused the variation in results. The concentrations of markers in urine and blood may reflect different biological processes, and this must be taken into account in the analysis.

R : We thank the reviewer for the suggestion. We agree that OPN, KIM-1, and Fetuin-A concentrations in urine and blood may reflect distinct biological processes, which could contribute to variability in the results. We also have added some points related to different samples and their relation to our result. OPN, KIM-1, and Fetuin-A exhibit different physiological roles depending on the sample type. For instance, urinary concentrations of KIM-1 primarily indicate tubular injury and localized renal processes, whereas its presence in blood may reflect systemic inflammation or broader kidney dysfunction (Han et al., 2008). To address this variation, we conducted subgroup analyses stratified by sample type where possible. This allows us to explore whether the type of sample influenced the prognostic utility of these markers. However, after conducting sensitivity analysis, either urine or blood samples provide a comprehensive view of kidney status. Studies suggest that both samples can be used independently or together to enhance the accuracy of ESRD prediction.

Reference :

• Han, W.K., Waikar, S.S., Johnson, A., Betensky, R.A., Dent, C.L., Devarajan, P. and Bonventre, J.V., 2008. Urinary biomarkers in the early diagnosis of acute kidney injury. Kidney international, 73(7), pp.863-869.

4. Uncertainty over cut-off values for clinical application: There is a problem with clinical application because there is no clear cut-off value for predicting the risk of developing ESRD based on the level of each biomarker. The cut-off value for biomarkers is an important indicator for predicting prognosis, so this point needs to be further investigated.

R : We appreciate the reviewer’s emphasis on the importance of establishing cut-off values for biomarkers in clinical practice. We acknowledge that determining precise cut-off values is critical for clinical application and prognosis prediction. However, we would like to clarify the limitations of the current study design and the availability of data in the included studies. Most of the included studies evaluated the association between biomarker levels and the risk of ESRD by comparing the mean concentrations of biomarkers, such as OPN, KIM-1, and Fetuin-A, between patients who progressed to ESRD and those who did not. This approach provides valuable insights into the general relationship between biomarker levels and ESRD risk but does not allow for the determination of specific thresholds at which risk significantly increases.

To define cut-off values, studies would need to stratify patients into multiple subgroups based on biomarker concentration intervals. For example, patients could be categorized into ranges of low, medium, and high biomarker levels, and the corresponding rates of ESRD within each group would need to be reported. Such data would allow researchers to identify the concentration level at which the risk of ESRD becomes significantly elevated, thus establishing a clinically actionable threshold. Unfortunately, studies employing this design are currently limited in the available literature. Most studies report mean differences or use continuous data without stratification, leaving a gap in the specific analysis required to determine cut-off values.

Additionally, there are inherent challenges in conducting such analyses. These include the need for large patient cohorts to ensure adequate representation across biomarker concentration intervals, as well as standardization in biomarker measurement methods and reporting. The lack of uniformity across studies further complicates efforts to extract or calculate meaningful cut-off values from existing data. Given these limitations, our study focuses on synthesizing the available evidence to highlight the general prognostic value of these biomarkers while acknowledging the need for future research specifically designed to address cut-off determination.

Reviewer 4 :

1. The manuscript titled "OSTEOPONTIN, KIDNEY INJURY MOLECULE-1, AND FETUIN-A AS PROGNOSTIC MARKERS OF END-STAGE RENAL DISEASE: A SYSTEMATIC REVIEW AND META-ANALYSIS" by Welliam et al., is clearly written and their claims are fully supported by well-designed study and proper statistical analysis.

R : We thank the reviewer for the positive feedbacks.

---

## [Decision Letter · Decision Letter 1]

25 Feb 2025

OSTEOPONTIN, KIDNEY INJURY MOLECULE-1, AND FETUIN-A AS PROGNOSTIC MARKERS OF END-STAGE RENAL DISEASE: A SYSTEMATIC REVIEW AND META-ANALYSIS

PONE-D-24-34763R1

Dear Dr. Wungu,

We’re pleased to inform you that your manuscript has been judged scientifically suitable for publication and will be formally accepted for publication once it meets all outstanding technical requirements.

Kind regards,

Wisit Kaewput, MD

Academic Editor

PLOS ONE

Additional Editor Comments (optional):

Accept as is.

Reviewers' comments:

Reviewer's Responses to Questions

**Comments to the Author**

1. If the authors have adequately addressed your comments raised in a previous round of review and you feel that this manuscript is now acceptable for publication, you may indicate that here to bypass the “Comments to the Author” section, enter your conflict of interest statement in the “Confidential to Editor” section, and submit your "Accept" recommendation.

Reviewer #1: All comments have been addressed

Reviewer #2: All comments have been addressed

2. Is the manuscript technically sound, and do the data support the conclusions?

Reviewer #1: Yes

Reviewer #2: Yes

3. Has the statistical analysis been performed appropriately and rigorously? 

Reviewer #1: Yes

Reviewer #2: Yes

4. Have the authors made all data underlying the findings in their manuscript fully available?

Reviewer #1: Yes

Reviewer #2: Yes

5. Is the manuscript presented in an intelligible fashion and written in standard English?

Reviewer #1: Yes

Reviewer #2: Yes

6. Review Comments to the Author

Reviewer #1: The manuscript looks good. I do not have any further comments and support the publication of the manuscript.

Reviewer #2: This systematic review & meta-analysis study has conducted according to Preferred Reporting Items for Systematic Reviews and Meta-Analysis (PRISMA) 2020 thoroughly, and thus meet the standard to be publish on PLOS One journal.

7. PLOS authors have the option to publish the peer review history of their article (what does this mean? ). If published, this will include your full peer review and any attached files.

**Do you want your identity to be public for this peer review?** For information about this choice, including consent withdrawal, please see our Privacy Policy .

Reviewer #1: No

Reviewer #2: **Yes: ** Fachira Rachel Agfata

---

## [Editor Report · Acceptance letter]

PONE-D-24-34763R1

PLOS ONE

Dear Dr. Wungu,

I'm pleased to inform you that your manuscript has been deemed suitable for publication in PLOS ONE. Congratulations! Your manuscript is now being handed over to our production team.

Kind regards,

on behalf of

Dr. Wisit Kaewput

Academic Editor

PLOS ONE